# chiLife: An open-source Python package for *in silico* spin labeling and integrative protein modeling

**Maxx H. Tessmer, Stefan Stoll** *

Department of Chemistry, University of Washington, Seattle, Washington United States of America

* stst@uw.edu

## Abstract

Here we introduce chiLife, a Python package for site-directed spin label (SDSL) modeling for electron paramagnetic resonance (EPR) spectroscopy, in particular double electron–electron resonance (DEER). It is based on *in silico* attachment of rotamer ensemble representations of spin labels to protein structures. chiLife enables the development of custom protein analysis and modeling pipelines using SDSL EPR experimental data. It allows the user to add custom spin labels, scoring functions and spin label modeling methods. chiLife is designed with integration into third-party software in mind, to take advantage of the diverse and rapidly expanding set of molecular modeling tools available with a Python interface. This article describes the main design principles of chiLife and presents a series of examples.

## Author summary

Thanks to modern modeling methods like AlphaFold2, RosettaFold, and ESMFold, high-resolution structural models of proteins are widely available. While these models offer insight into the structure and function of biomedically and technologically significant proteins, most of them are not experimentally validated. Furthermore, many proteins exhibit functionally important conformational flexibility that is not captured by these models. Site-directed spin labeling (SDSL) electron paramagnetic resonance (EPR) spectroscopy is a powerful tool for probing protein structure and conformational heterogeneity, making it ideal for validating, refining, and expanding protein models. To extract quantitative protein backbone information from experimental SDSL EPR data, accurate modeling methods are needed. For this purpose, we introduce chiLife, an open-source Python package for SDSL modeling designed to be extensible and integrable with other Python-based protein modeling packages. With chiLife, appropriate SDSL EPR experiments for protein model validation can be designed, and protein models can be refined using experimental SDSL EPR data as constraints.

**Data Availability Statement:** chiLife is a free and open-source python API and is available from https://github.com/StollLab/chiLife. A stable version of chiLife can be installed using the pip package manager by running pip install chiLife. A

development version can be installed from GitHub using the instructions available on the GitHub page.

**Funding:** This work was supported by the National Institutes of Health (NIH), grant R01 GM125753 (S.S.). The spectrometer used was funded by NIH grant S10 OD021557 (S.S.). The funders had no role in study design, data collection and analysis, decision to publish, or preparation of the manuscript.

**Competing interests:** The authors have declared that there are no competing interests.

This is a *PLOS Computational Biology* Software paper.

## 1. Introduction

Site-directed spin labeling (SDSL) electron paramagnetic resonance (EPR) is a powerful method for investigating protein structure and dynamics [1–3]. Solution continuous-wave (CW) EPR probes global motions, like protein tumbling, and local motions, like side chain and backbone dynamics [4,5], which provide valuable information on protein topology, tertiary and quaternary structure, and functionally important protein dynamics. Power saturation EPR measures spin label solvent accessibility and membrane depth [6,7], which can provide information on protein transmembrane insertion and topology and inform protein–membrane docking of peripheral membrane proteins [8–10]. Pulse dipolar EPR experiments such as double electron–electron resonance (DEER) determine distance distributions between pairs of spin labels [11–15]. These distance distributions directly provide information on protein tertiary and quaternary structure. Coupled with high-resolution structural techniques, SDSL EPR provides insight into protein conformational landscapes and how they change in response to different stimuli [16–27]. Accordingly, SDSL EPR is particularly useful for validation and refinement of protein structural models as well as expansion of these models to include conformational heterogeneity and distinct alternate conformational states. SDSL EPR has been used to validate, refine, and expand upon models developed using experimental structure determination methods such as x-ray crystallography and cryo-electron microscopy. Now, with modern computational methods like AlphaFold2, RosettaFold and ESMFold [28–31], SDSL EPR have become even more valuable for these tasks.

SDSL EPR is predominantly performed by introducing one or more cysteines via site-directed mutagenesis and attaching a thiol-reactive spin label reagent such as S-(1-oxyl-2,2,5,5-tetramethyl-2,5-dihydro-1H-pyrrol-3-yl) methyl methanesulfonothioate (MTSL), yielding the spin-labeled side chain R1 [5]. While R1 is the most popular spin label, several alternatives exist that offer different reaction chemistries, linker lengths, chemical stability, and other properties that may be desirable depending on the experiment [32,33]. All data gathered from SDSL EPR experiments are necessarily co-determined by the spin label structure, dynamics, and environment in addition to the structure and dynamics of the protein they are attached to. Therefore, to obtain quantitative information about protein structure and dynamics from SDSL EPR data, it is crucial to accurately model the local structure and dynamics of the spin label.

To date, several spin label and protein modeling applications have been developed to aid in experimental design, interpretation, and protein modeling with SDSL EPR data [34–49]. While these methods generally perform well, it is currently difficult to use them with novel protein modeling protocols, integrate them with existing protein modeling software, or utilize them with novel spin labels. Several of these applications only allow for prediction of distance distributions and cannot be used to predict other types of experimental results such as membrane depth or solvent accessibility [37,40,43,50]. Of the available software that predict distance distributions, only a small number provide predefined docking or conformational change algorithms [34,45–48] that can only be minimally altered by the user. Integration with third-party modeling applications is often severely limited due to a lack of a scriptable interface. Additionally, most of these packages only implement one or a few spin labels and do not offer the ability to add new spin labels easily. Recently, significant efforts have been made to make spin label modeling more integrable and scriptable [49]. A spin label modeling package that integrates well with other protein modeling packages and allows users to easily add their

own spin labels would aid investigators in the development of novel modeling protocols and utilization of cutting-edge protein modeling methods with SDSL restraints. These advancements would aid development, validation, and refinement of protein models, as well as the ability to expand these models to include alternate conformational states.

Here we introduce chiLife, a scriptable SDSL modeling package designed as a tool to develop novel SDSL EPR modeling and analysis pipelines. chiLife models spin labels on proteins, providing direct access to all the methods and properties of the spin labels, as well as allowing users to easily implement custom spin labels. chiLife is written in Python and thus can be integrated with the wide variety of Python-based protein modeling and analysis applications, such as MDAnalysis [51], PyRosetta [52], and Xplor-NIH [53,54]. In addition to EPR applications, chiLife can be used for other experiments such as paramagnetic relaxation enhancement (PRE) [55,56] or electron–nuclear double resonance (ENDOR) [57]. Below we provide an overview of the core functionality of chiLife and demonstrate several possible use cases through examples.

## 2. Design and implementation

The central entities in chiLife are `SpinLabel` objects (Fig 1). A `SpinLabel` is derived from the parent `RotamerEnsemble` object, which represents a weighted ensemble of side chain

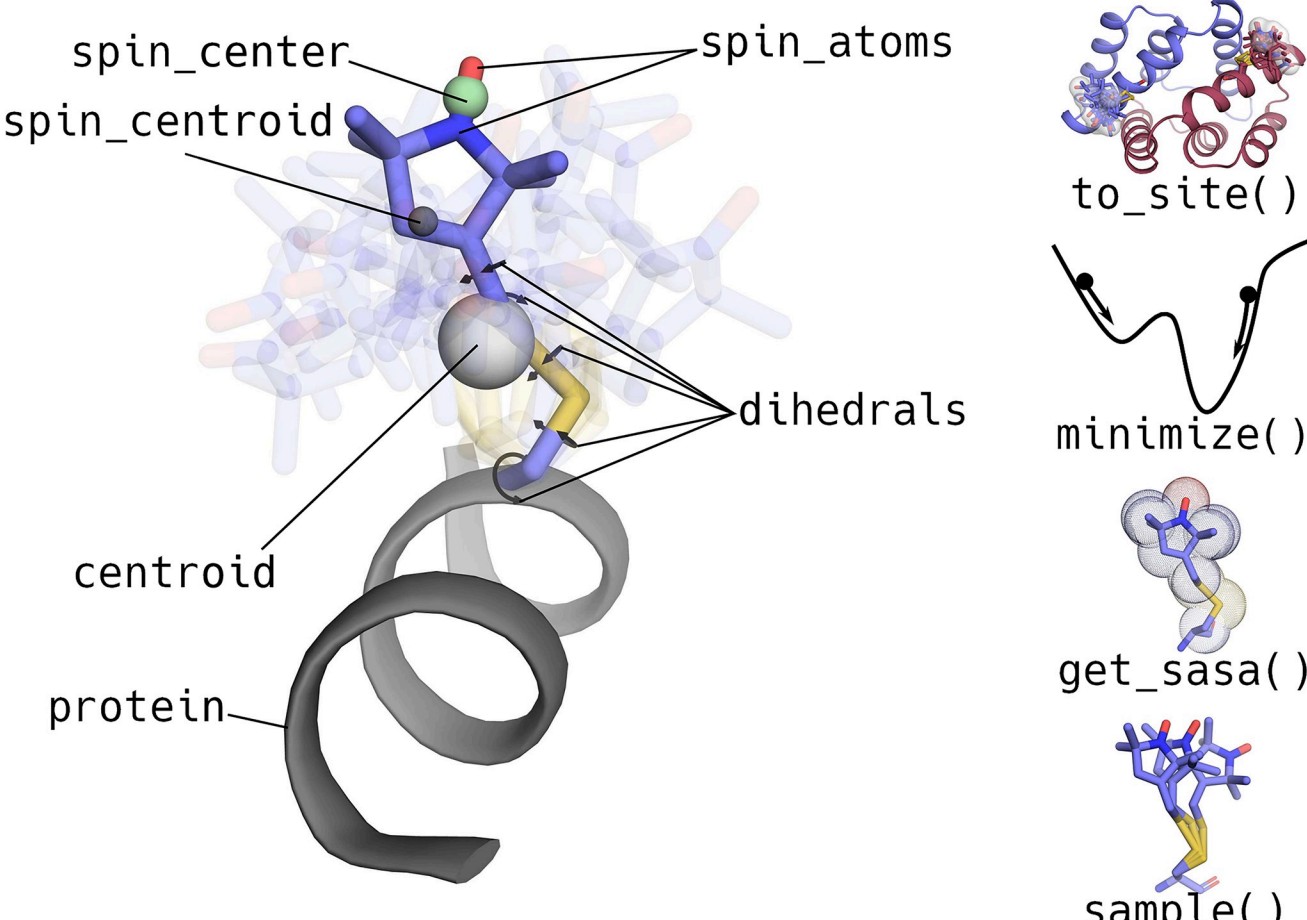

**Fig 1. Illustration of chiLife's `SpinLabel` object.** On the left, some of its user-accessible properties are shown. On the right, some useful methods are illustrated that allow users to modify or calculate new structures from the `SpinLabel` object.

rotamers aligned to a backbone site. Rotamers are represented by sets of values of their mobile dihedral angles (`dihedrals`), associated with weights (`weights`). Each dihedral angle can optionally be normally distributed, with separate standard deviations (`dihedral_sigmas`). Bond lengths and bond angles are fixed. The `SpinLabel` object extends the `RotamerEnsemble` object by adding the unpaired electron spin density, which is represented as a weighted distribution over one or more spin-bearing atoms (`spin_atoms`, `spin_weights`, `spin_coords`). This allows chiLife to represent labels with fully localized spin density (such as trityls, $Cu^{2+}$, $Gd^{3+}$) as well as with small delocalization (nitroxides) and extensive delocalization (triplet states of porphyrins [58]). The centers of the delocalized spin density of all rotamers are obtainable by the `spin_centers` property. These properties, along with many others, can be used to simulate experimental observables, validate and restrain computational models, and aid in the development of efficient and accurate experimental designs.

A `SpinLabel` object is usually created in the context of a protein by loading a rotamer library and attaching it to a protein structure. To interact with protein models, chiLife uses the widely used MDAnalysis library [51]. This allows users to make use of the rich features and atom selection language offered by MDAnalysis. A `SpinLabel` is attached to a target protein site by aligning the rotamers from the rotamer library and target site by a method called bisect. This method translates the rotamers to superimpose their CA atoms with the CA of the protein and rotates the rotamers such that the N–CA–C planes and the vectors bisecting the N–CA–C angles are aligned. This alignment method assures that the side chain atoms are not biased to one side of the residue in the case that the rotamer library and the protein target site have different N–CA–C angles. Other alignment methods are implemented and selectable during `SpinLabel` construction. If the `sample = n` argument is given, where n is a user-defined number of samples, a set of off-rotamer samples [59] is generated and attached instead of the rotamer library. The degree of off-rotamer sampling is controlled by the rotamer library being used but can be overridden by the `dihedral_sigmas` keyword argument. If `dihedral_sigmas` is set to `numpy.inf` then the sampling method effectively samples the entire volume accessible to the spin label side chain [36,45,46]. Next, clashes between the sampled rotamers and surrounding side chains are evaluated using a flat-top repulsive Lennard–Jones potential energy function, and rotamers with high predicted total energy and consequently low population are trimmed from the ensemble. A forgive factor for the Lennard–Jones potential as well as a maximum distance for clash evaluations can be provided. The energy function used can be modified by providing an alternate built-in or a user-defined energy function via the `energy_func` keyword argument. Clash evaluations can be turned off via the `eval_clash` keyword argument.

Once attached, a spin label can be further manipulated or used for analysis as illustrated in Fig 1. Summary quantities such as `centroid`, the centroid of all the heavy atoms of the attached ensemble, and `spin_centroid`, the centroid of the `spin_centers` of all rotamers in the ensemble, can be used when modeling systems with aggregate measurements like membrane depth. The `SpinLabel` object gives users control over how rotamer ensembles are constructed, how energies are evaluated, and how the ensembles are manipulated, while offering practical defaults and access to attributes that can be used to predict experimental observables.

## 3. Results

### 3.1. Basic spin label modeling

chiLife has a range of built-in methods for modeling spin labels on proteins. The first example illustrates three of them: the rotamer library (RL) method [34,41], used if the `sample`

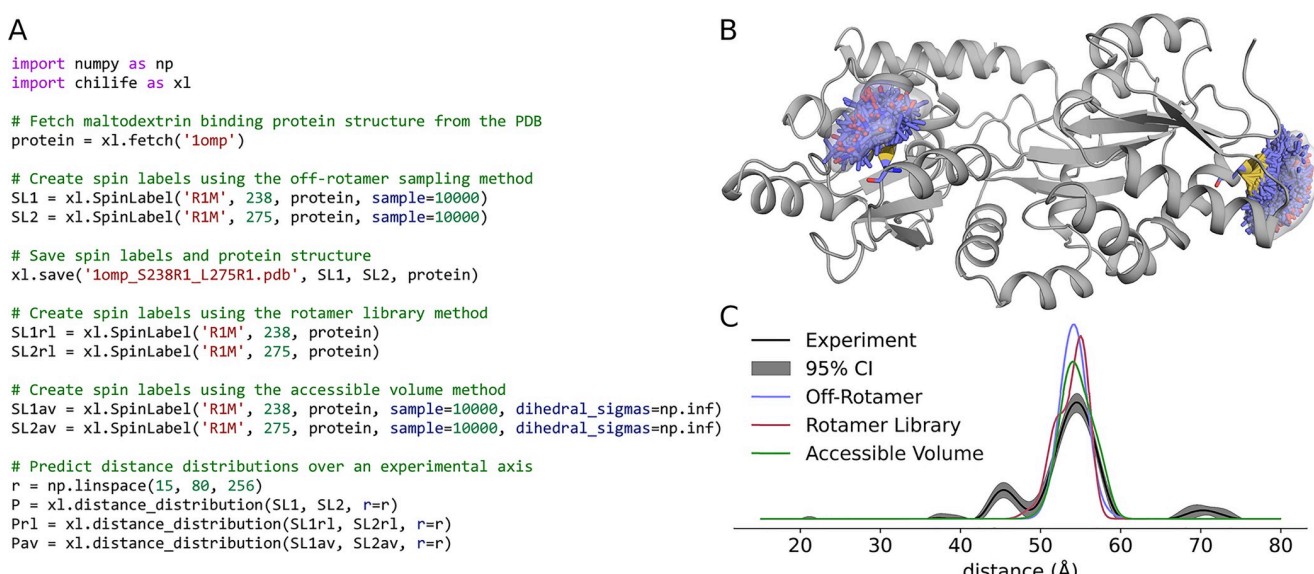

**Fig 2. Spin labeling proteins and predicting spin label distance distributions.** A) chiLife script. B) Cartoon model of maltose binding protein (PDBID 1OMP) labeled with R1 at sites 238 and 275, showing the spin label ensembles (sticks) and weighted kernel density estimates of the spin centers (semitransparent surfaces). C) Comparison of the predicted distance distributions with the experimental distance distribution.

keyword argument is not set, the accessible-volume (AV) method [36,45], accessed by setting `sample` to an integer and `dihedral_sigmas` to `numpy.inf`, and an off-rotamer sampling method [59] that is controlled by sample and `dihedral_sigmas` arguments. These methods are illustrated in the script shown in Fig 2A. In this example, a protein object is created by fetching a structure of maltodextrin/maltose binding protein (MBP) from the PDB [60]. The fetch function returns an MDAnaysis `Universe` object which contains all chains, states, and ligands of the fetched PDB. Users can utilize the MDAnalysis `select_atoms` function to create an `AtomGroup` containing only the chains, ligands, or atoms of interest [51]. When provided with an `AtomGroup`, chiLife will only consider these atoms when evaluating external interactions, except for water molecules which are implicitly excluded. The exclusion of any other atoms must be managed by the user. Next, `SpinLabel` objects are constructed for two sites using the three different methods described above. The first two spin label ensembles and the protein structure are saved to a new PDB file for visualization. Finally, distance distributions are generated from each pair of `SpinLabel` objects using the `distance_distribution` function. This function accepts an arbitrary number of spin labels and returns the cumulative distribution from all spin label pairs. The RL and AV methods were initially implemented in the applications MMM [34,41] and MTSSLWizard [36,45] respectively, and while chiLife supports them, most default parameters are not the same. Thus, to be able to accurately replicate results from these packages, chiLife offers the `from_mmm` and `from_wizard` class methods.

Fig 2B plots the spin label objects as stored in the PDB file. This file contains an unaltered copy of the protein (if provided), all the spin labels as separate multistate models with their relative populations stored as occupancy factors, and a set of pseudo-atom coordinates representing the spin centers of the spin label rotamers. These pseudo-atoms are shown as a surface in Fig 2B with the relative populations mapped to the color of the surface. Fig 2C shows the distance distributions predicted by the three labeling methods and compares them to the experimentally determined distribution [59]. All three methods provide consistent prediction of a predominant

distance at about 55 Å. The smaller peak at about 45 Å likely is due to a subpopulation of MBP in the maltose-bound conformation that is known to be sampled in the absence of maltose [56].

## 3.2. SDSL screen of solvent-accessible surface residues

One important application of *in silico* spin label modeling is the use of site pair scans to predict distance distributions for designing experiments that test protein models. With chiLife, a script can be used for screening solvent-exposed site pairs of proteins in order to find the optimal spin labeling sites to investigate conformational changes or to obtain experimental evidence that best discerns between competing protein models. Listing S1 in S1 Text performs a screen of all solvent-accessible surface residues (>50 Å$^2$ solvent accessible surface) of MBP in the apo and holo state for maximal distance contrast. This is done by modeling spin labels at all sites and predicting pairwise distance distributions between all labels for both states. Then the predicted distributions are screened for site pairs that distinguish the bound (PDBID: 1ANF) and unbound (PDBID: 1OMP) states of MBP, in this case by maximizing the earth-mover's distance between the two distributions.

Fig 3A shows the spin labels of the best site pair (residues 38 and 352) attached to the two conformations of MBP and Fig 3B shows the two predicted distance distributions. This type of analysis facilitates designing the optimal SDSL site pairs when investigating protein conformational change. In some cases, this may be superior to screening for a change in protein backbone distance alone since it considers the relative orientation and rotamer distribution of the spin labels. To illustrate this, Fig 3C and 3D show a site pair (residues 45 and 211) with a significant change in the backbone Cβ–Cβ distance, but little change in the spin label distance distribution.

## 3.3. Adding custom spin labels

In addition to chiLife's built-in rotamer libraries for common spin labels, users can add rotamer libraries for new spin labels. This is important as new spin labels are continuously being developed for various applications featuring bio-orthogonal coupling chemistry [61–63], shorter linker lengths [64,65], enhanced phase memory times [66], and resistance to chemical reduction [67]. Listing S2 in S1 Text creates new chiLife rotamer libraries for the three spin labels R3A [68], Gd(III)-DO3A [69] and NOBA [70] using multistate PDB files. Fig 4 illustrates the use of the new rotamer libraries on T4 lysozyme.

The rotamer libraries consist of a set of low-energy conformers and their associated internal energies generated using CREST [71]. Each new rotamer library is created using the `create_library` function, which allows the user to specify the mobile dihedral angles as a list of quadruplets, and the atoms and fractional populations for the unpaired electron spin density. The rotamers of the library can also be weighted by providing an array or list of weights via the `weights` argument.

The `create_library` function is very versatile and can create libraries from a single-state or multi-state PDB file. It can create libraries that contain only one structure that can randomly sample new dihedral angles, and libraries that have multiple rotameric states but no independent rotatable side chain dihedral angles such as 2,2,6,6-tetramethyl-N-oxyl-4-amino-4-carboxylic acid (TOAC). Additionally, when using a multi-state PDB file, chiLife will retain any stereoisomeric heterogeneity present among the states in the file. This feature is particularly useful for labels with reaction chemistries which create diastereomers when reacting with thiols, such as maleimides.

The output of the `create_library` function is a file in NPZ format which can be used by specifying the `rotlib` keyword argument when constructing a `SpinLabel`. These

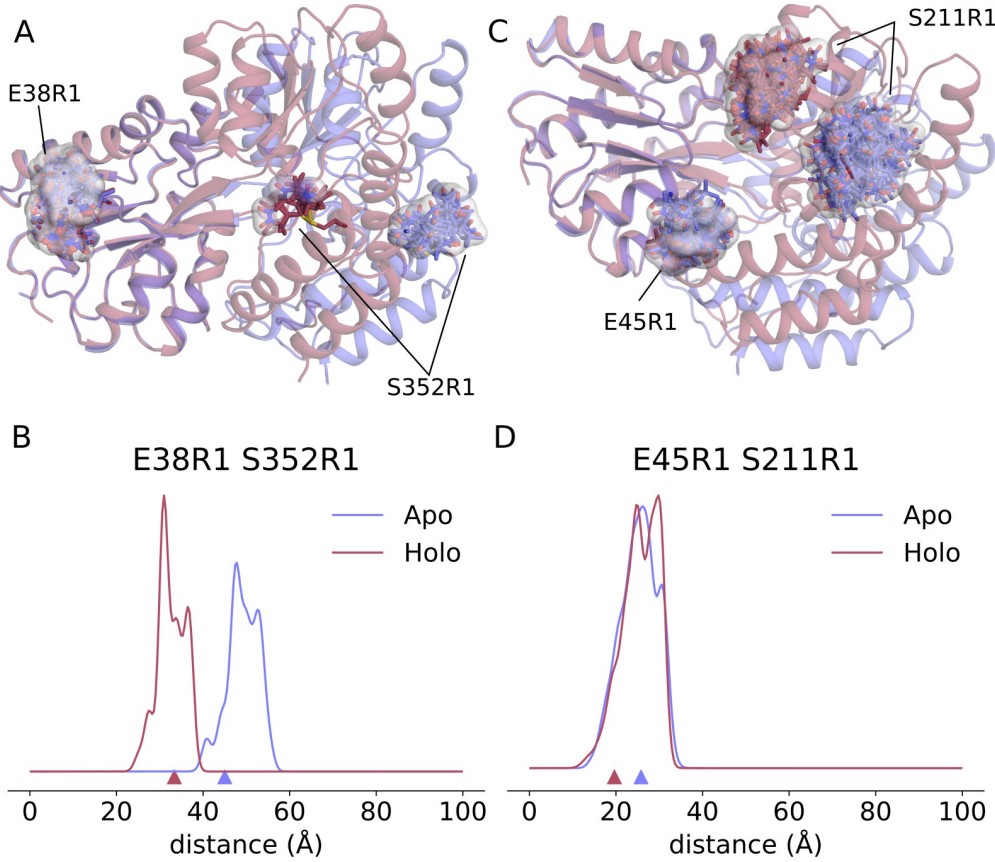

**Fig 3. Illustration of how changes in backbone do not necessarily cause changes in distance distributions.** Top: Comparison of apo (blue, PDBID 1OMP) and holo (red, PDBID 1ANF) MBP structures and the locations of the model R1 spin labels (sticks) for the site pairs E38R1 S352R1 (A) and E45R1 S211R1 (B). Bottom: Comparison of apo and holo distance distributions for the two site pairs that both show significant changes in Cβ–Cβ backbone distance, indicated by small triangles at the base of the plots. The E38R1 S352R1 site pair on the left (C) shows a clear difference in the predicted distributions while the E45R1 S211R1 site pair on the right (D) shows very little change.

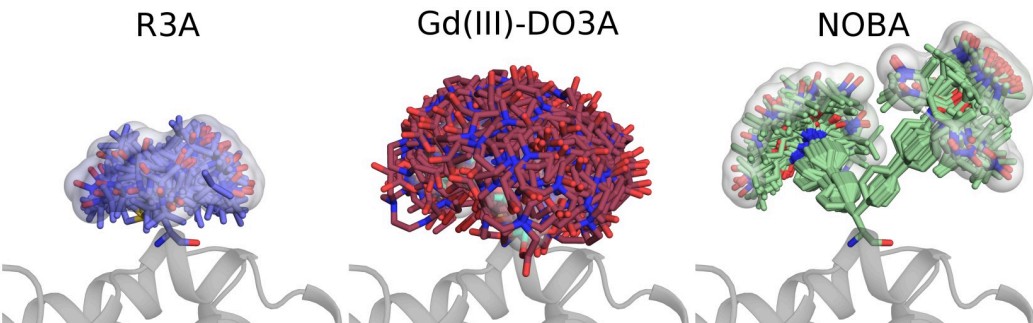

**Fig 4. Three spin labels added to chiLife and attached to T4 lysozyme (PDBID: 2LZM) at site 109.** R3 (left, sticks with blue carbons) is a small highly mobile nitroxide label. Gd(III)-DO3A (center, sticks with dark red carbons) is a gadolinium-based spin label resistant to reduction. NOBA (right, sticks with green carbons) is a biorthogonal nitroxide. The surface is made by pseudo-atoms at the rotamer spin centers.

rotamer library files can be shared with coworkers and collaborators. The rotamer library name is distinct from the residue name, allowing the use of multiple rotamer libraries for the same residue type, which can aid in rotamer library development and benchmarking.

### 3.4. Local side chain repacking

When modeling spin labels very close together or in crowded environments, it might be important to model changes in the conformations of neighboring side chains as well by performing local side chain repacking. This can be accomplished using the chiLife `repack` function, as illustrated in Listing S3 in S1 Text and Fig 5. The `repack` function uses Markov chain Monte Carlo (MCMC) sampling to repack [72] all residues within a user-defined radius of a set of `SpinLabel` objects. To do this, chiLife relies on the widely used Dunbrack rotamer libraries [73] for canonical amino acid side chains. In each MCMC sampling step, a spin label or neighboring site is chosen at random and a new rotamer is sampled from the rotamer library associated with that site. If the `off_rotamer` option is set to `True`, new off-rotamer dihedrals are sampled for that rotamer. The step is accepted or rejected using the Metropolis–Hastings criterion based on energy. The outputs of the repack function are an MDAnalysis `Universe` object of the protein structures for all the accepted steps and a list of the relative energies for all steps. From this trajectory, the `from_trajectory` method builds a new `SpinLabel` object (neglecting a user-adjustable number of initial burn-in steps), which can then be used in the same way as any other `SpinLabel` object in chiLife.

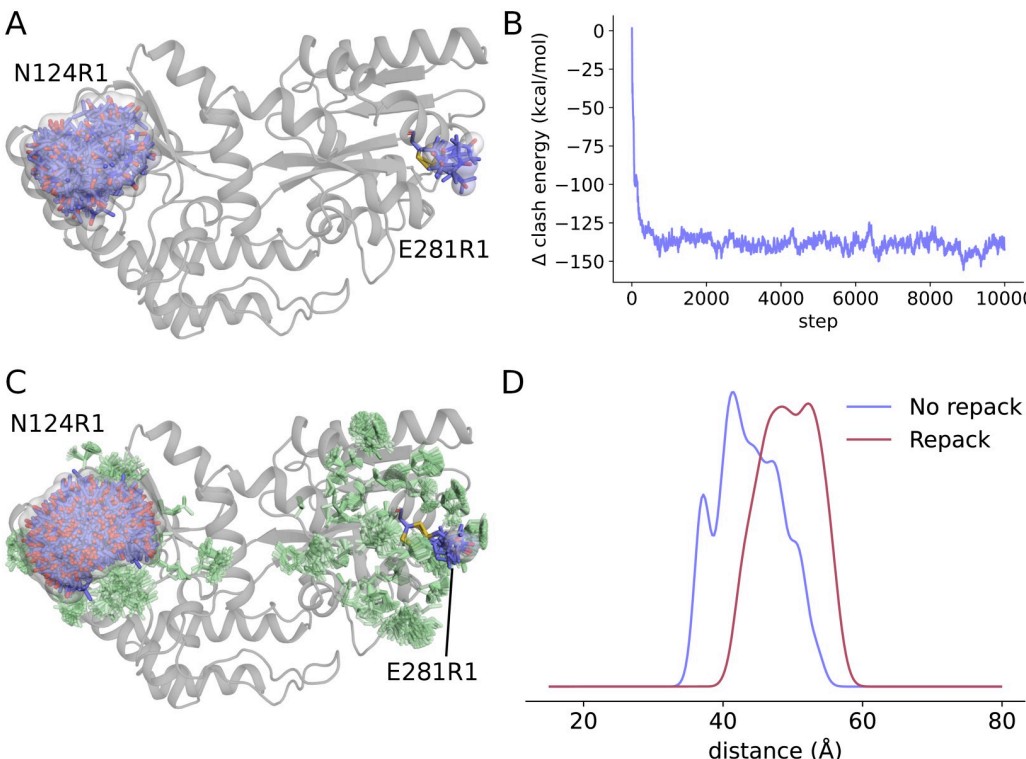

**Fig 5. Local side chain repacking for R1 spin label ensembles at sites N124R1 and E281R1 on MBP (PDBID 1OMP).** A) Prediction of spin label ensembles without repacking. The protein structure is shown as a gray cartoon. Spin labels are shown as blue sticks. B) Energy trajectory of MCMC repacking, relative to the energy of the starting structure. C) Spin label (blue sticks) and neighboring side chain (green sticks) ensembles obtained from the repacking trajectory. D) Comparison of predicted distance distributions derived from the ensembles of the repacked and the original structures.

Fig 5 illustrates the effect, and potential benefit, of repacking. Fig 5A shows MBP labeled at two sites using the ORS method. Despite ORS, the rotamer ensembles are still confined by the rigid neighboring residues. Fig 5B shows the trajectory of the energy score function during MCMC repacking. It illustrates a rapid improvement in energy during the first 1000 steps, followed by a long steady-state sampling of spin label and neighboring amino acid side chain conformations. Fig 5C shows the results of this repacking, and Fig 5D plots the predicted distance distributions derived from the original and repacked structures. The repacked structure shows a distance distribution that is significantly different from the one made without repacking.

### 3.5. Membrane docking

While EPR excels at measuring ensemble distance distributions between spin labels, it is also useful for providing data to answer other questions such as how and where proteins interact with membranes [6]. With chiLife, users can accomplish modeling tasks driven by many types of SDSL EPR data. As an example, Listing S4 in S1 Text utilizes spin label membrane depth data from Malmberg et al. [8] to determine the position and orientation of the C2 domain of cytosolic phospholipase A2 (cPLA2) in the membrane. These data are calculated from power saturation EPR measurements of spin labels on several sites in the presence of relaxation agents with different membrane permeability. For each site, a `SpinLabel` object is created, and its `spin_centroid`, the weighted average position of the `spin_atoms` over the whole rotamer ensemble, is determined. Then, a least-squares fit is performed to determine the Z-position of the protein and the three Euler angles describing its orientation by minimizing the difference between the Z-coordinates of the centroid and the experimentally determined depth. The resulting model, shown in Fig 6, is in good agreement with previously reported models [8,74]. This approach takes full advantage of the spin label modeling methods available in chiLife and does not require manual modeling of rigid rotamers as performed previously [8,74], nor does it require mutating the original amino acids. Similar protocols can be

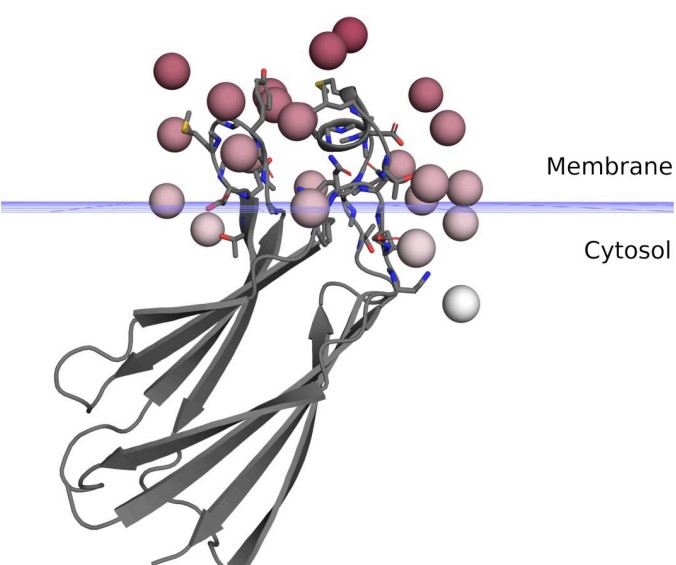

**Fig 6. Membrane docking of the cPLA2 C2 domain (PDBID 1BCI).** The cPLA2 C2 domain is shown as a gray cartoon. Spin centroids are shown as red spheres with their color saturation mapped to the experimental depth. Native side chains of spin labeled sites are shown as sticks. The blue grid indicates the phosphate plane of the model.

developed to dock or orient other membrane-associated proteins in a lipid bilayer. Additionally, modelling protocols for other SDSL EPR experiments can be developed making use of solvent accessibility and spin label mobility data [75,76].

### 3.6. Custom scoring functions

In addition to custom modeling protocols, chiLife allows users to define and utilize their own energy functions, either independently developed or interfaced from other molecular modeling methods. The only requirement for an energy function is that it accepts a `RotamerEnsemble` or `SpinLabel` object and outputs an energy for each rotamer in the ensemble in kcal/mol. Fig 7 provides an example where the scoring function, consisting of a modified Lennard–Jones potential with a forgive factor and a maximum energy cap, is augmented with an additional attractive term proportional to the solvent accessible surface area of the rotamer. This term is meant to capture the van der Waals forces between the solvent and solvent-accessible surface atoms of the rotamers. The motivation for this is that the attractive force contributions of the Lennard–Jones potential can bias the rotamers to form intramolecular interactions with the surface of the protein if the compensatory van der Waals interactions with solvent molecules are neglected [77]. The forgive factor and the weight of the SASA term were fitted to produce the best agreement with a recently published DEER data set for MBP [59]. Fig 7 shows how this custom scoring function performs on two previously published constructs of MBP [59] which use highly solvent-accessible sites. For these data, the SASA-augmented score function results in significant improvements in the distance distribution prediction accuracy compared to the modified Lennard–Jones potential alone.

While this example illustrates the customizability of chiLife, it also illustrates how chiLife can be used in an exploratory fashion to further develop spin label modeling methods. Replacing the custom score function with methods from third-party software allows to integrate novel developments in other fields into spin label modeling. For example, use of the general forcefield offered by xTB has recently shown promise as a powerful tool for modeling spin labels, because of its high-resolution score function and its rapid and accurate parametrization of spin labels, which are often difficult to parametrize for traditional force fields [78,79]. Similarly, deep-learning potentials [80,81] show promise as high-resolution score functions for spin label modeling.

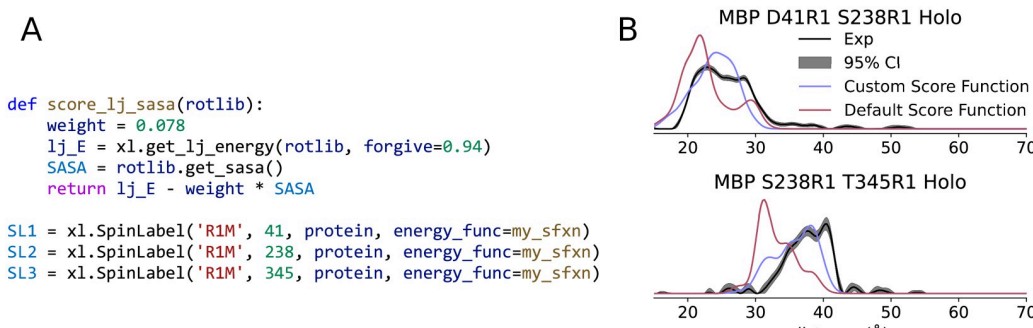

**Fig 7. Comparison of distance distribution predictions using different scoring functions.** A) Definition and application of custom energy function using chiLife. B) Experimental distance distributions for two site pairs of MPB, taken from [59], are compared to the predicted distance distributions from spin labels modeled using a modified Lennard–Jones potential and a custom function that augments the same potential with an additional term to account for compensatory attractive forces with the solvent.

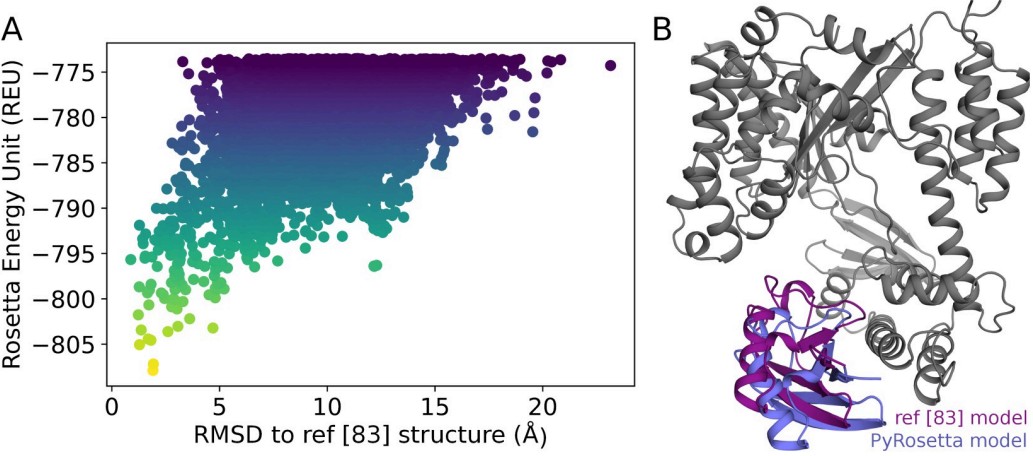

**Fig 8. Comparison of a previously published ExoU–ubiquitin complex model and the best scoring model obtained by integrating chiLife and PyRosetta.** A) Docking funnel showing convergence towards the previously published complex using chiLife restraints. B) Cartoon structures of ExoU (gray) showing predicted locations of ubiquitin in the previously published model (purple) and in the PyRosetta model produced here (blue).

## 3.7. Interaction with third-party software: Rosetta

Because chiLife is written in Python and designed for modular modeling pipelines, it can be integrated with other Python-based molecular modeling packages. This integration can be used to aid chiLife in modeling spin labels as discussed above, or to use chiLife to aid protein modeling. Listing S5 in S1 Text is a PyRosetta [52] script illustrating how chiLife can be integrated with PyRosetta. Using previously published DEER data obtained for nine site pairs [82,83], it models the interaction between the bacterial protein toxin ExoU and its cofactor ubiquitin [82–84]. Fig 8 compares the model obtained using PyRosetta and chiLife with the previously published and experimentally validated model [83]. Fig 8A shows docking funnels that confirm convergence of the algorithm towards the previously published structure. Fig 8B compares the two models [83]. Both models have a hydrophobic patch of ubiquitin interfacing with the same hydrophobic residues of ExoU α-helix 18, with a Cα root mean square deviation (RMSD) of less than 2 Å.

This example illustrates how chiLife can be integrated with other protein modeling Python packages. Notably, Rosetta has some built-in utilities for modeling proteins with the R1 spin label [38,39]; however, the data used here include some DEER experiments conducted with the brominated spin label R7, which produces different distance distributions than R1 [85]. While R1 only differs from R7 by one atom (H vs. Br), this approach is readily applicable to all spin labels supported by chiLife as well as custom spin labels. Additionally, any of the previously published spin label modeling methods can be used, including the rotamer library approach [41], the accessible volume approach [36] and the off-rotamer sampling approach [59]. Because of the diversity and modularity of PyRosetta, the scoring term used in this example can also be used to perform other modeling tasks such as structural refinement, *ab initio* folding, or conformational change modeling.

## 4. Availability and future directions

SDSL EPR is a powerful integrative method for probing protein structure and dynamics. To aid these investigations, we have developed chiLife, a modular, scriptable spin label modeling

package that facilitates the rapid development of application-specific protein modeling pipelines using SDSL EPR data. We described several examples that illustrate how chiLife can be used for experimental design, analysis, and protein modeling. As protein modeling methods are outpacing experimental structure determinations, the utility of integrative methods like SDSL EPR will become invaluable for model validation, refinement, and hypothesis development. chiLife will support these endeavors by offering flexible spin label modeling methods that can be integrated into custom modeling or analysis workflows.

chiLife is a free and open-source Python API and is available from https://github.com/StollLab/chiLife. A stable version of chiLife can be installed using the pip package manager by running pip install chiLife. A development version can be installed from GitHub using the instructions available on the GitHub page.

## Supporting information

**S1 Text. Supplementary Text.**
(PDF)

## Acknowledgments

We thank Eric G. B. Evans for providing feedback that helped improve several design aspects of chiLife.

## Author Contributions

**Conceptualization:** Maxx H. Tessmer, Stefan Stoll.

**Funding acquisition:** Stefan Stoll.

**Investigation:** Maxx H. Tessmer.

**Methodology:** Maxx H. Tessmer, Stefan Stoll.

**Resources:** Stefan Stoll.

**Software:** Maxx H. Tessmer.

**Supervision:** Stefan Stoll.

**Visualization:** Maxx H. Tessmer.

**Writing – original draft:** Maxx H. Tessmer.

**Writing – review & editing:** Maxx H. Tessmer, Stefan Stoll.

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
