## [Decision Letter · Decision Letter 0]

17 Feb 2023

Dear Prof. Stoll,

Thank you very much for submitting your manuscript "chiLife: An open-source Python package for in silico spin labeling and integrative protein modeling" for consideration at PLOS Computational Biology. As with all papers reviewed by the journal, your manuscript was reviewed by members of the editorial board and by several independent reviewers. The reviewers appreciated the attention to an important topic. Based on the reviews, we are likely to accept this manuscript for publication, providing that you modify the manuscript according to the review recommendations.

Sincerely,

Alexander MacKerell

Academic Editor

PLOS Computational Biology

Arne Elofsson

Section Editor

PLOS Computational Biology

Reviewer's Responses to Questions

**Comments to the Authors:**

Reviewer #1: In their article "chiLife: An open-source Python package for in silico spin labelling and integrative protein modelling", Tessmer and Stoll present a very nice piece of software that can be easily integrated into other Python code, allowing programmers to easily attach ensembles of spin labels to protein models and perform a number of calculations with such models.

The package is well designed, open source and the description in the paper is very good. The authors have included a number of examples and code listings which will make it easy for users to learn how to use the library.

This library will be an important addition to the field of spin-labelling EPR and I think, due to the way it is constructed, it will also be very useful in other fields such as FRET.

It would be great if the package could also work with nucleic acids. I guess, such a feature is already on the list for an upcoming version.

I recommend publication of this paper.

Some minor points:

1.It is not entirely clear to me how the fetch method deals with models that have multiple copies in the ASU - is it possible to fetch a specific chain of the model, e.g. using xl.fetch('1omp_A') or something similar? Will the user be warned that there are multiple chains?

2.PDB files sometimes contain ligands and solvent molecules that can interfere with calculations but are difficult to detect automatically. Sometimes it makes sense to include them, sometimes not. It would be great if the program could output a stripped-down version of the input .pdb file, containing all the atoms used in the calculations. Is such a method already included?

3.Are bipedal spin labels such as Rx or the Cu-NTA label supported?

Reviewer #2: In the article 'chiLife: An open-source Python package for in silico spin labeling and integrative protein modeling' the authors publish an truly useful follow-up article to their landmark paper 'Comparative evaluation of spin-label modeling methods for protein structural studies'. In the article the authors introduce and further provide detailed insights to chiLife package. The chiLife package has the capability to correctly model paramagnetic labels on biomolecuels that eventually lead information such as their distance between them and solvent accessibility. However the most striking ability is that it is written fully in python that makes it freely accessible and foremost easily integrated into other programs such as NIH-Xplor, pyRosetta and others. This is a big step forward to programs such as DeerRosetta and MMM.

The authors lead the reader through the article with numerous examples that become widely accessible to the none-EPR expert. Most important the reader can subsequently integrate chiLife in his/her own research. I deeply enjoyed and appreciated the overlap between the experimental and simulated distances presented in figure 1 and 7, that show the correct modeling approaches used in the publication. I appreciate the ease on how to add costume labels to chiLife that will become more useful in the timeline of bio-molecular EPR. Once labels are added to the biomolecule the local environment can be repacked via customized scoring function. Now while repacking has been applied to many systems before, here the authors introduce a 'scoring function', which gives the user the ultimate freedom to apply to wide range of systems that stretch further than the normal protein system. However I would like to have more information on how to apply those scoring function.

Lastly the authors treat an under estimated strength of EPR, which is the membrane docking function and solvent accessibility. Since the age of Hubell and side directed spin labeling researchers have used EPR to get information on localized environment that changes upon applying molecular stimuli. Specifically a study by Ralf Langen treated the membrane remodeling ability introduced by the bar domain. Here, we have a tool that can correctly interpret such data and see domain reorientation that introduce specific biological functions.

Overall, I want to recommend the article for publication. One slight addition would be a specific outline on how to install the software, which is in detail provided on the chiLife website with all other tutorials (see 'Availability').

**Have the authors made all data and (if applicable) computational code underlying the findings in their manuscript fully available?**

Reviewer #1: Yes

Reviewer #2: Yes

PLOS authors have the option to publish the peer review history of their article (what does this mean?). If published, this will include your full peer review and any attached files.

Reviewer #1: No

Reviewer #2: **Yes: **Thomas Schmidt

Figure Files:

Data Requirements:

Reproducibility:

References:

---

## [Editor Report · Decision Letter 1]

27 Feb 2023

Dear Prof. Stoll,

Thank you very much for submitting your manuscript "chiLife: An open-source Python package for *in silico* spin labeling and integrative protein modeling" for consideration at PLOS Computational Biology. As with all papers reviewed by the journal, your manuscript was reviewed by members of the editorial board and by several independent reviewers. The reviewers appreciated the attention to an important topic. Based on the reviews, we are likely to accept this manuscript for publication, providing that you modify the manuscript according to the review recommendations. Please address the minor criticisms of the reviewers and we can move the manuscript to full acceptance. 

Sincerely,

Alexander MacKerell

Academic Editor

PLOS Computational Biology

Arne Elofsson

Section Editor

PLOS Computational Biology

Figure Files:

Data Requirements:

Reproducibility:

References:

---

## [Editor Report · Decision Letter 2]

16 Mar 2023

Dear Prof. Stoll,

We are pleased to inform you that your manuscript 'chiLife: An open-source Python package for *in silico* spin labeling and integrative protein modeling' has been provisionally accepted for publication in PLOS Computational Biology.

Best regards,

Alexander MacKerell

Academic Editor

PLOS Computational Biology

Arne Elofsson

Section Editor

PLOS Computational Biology

---

## [Editor Report · Acceptance letter]

28 Mar 2023

PCOMPBIOL-D-22-01886R2 

chiLife: An open-source Python package for *in silico* spin labeling and integrative protein modeling

Dear Dr Stoll,

I am pleased to inform you that your manuscript has been formally accepted for publication in PLOS Computational Biology. Your manuscript is now with our production department and you will be notified of the publication date in due course.

With kind regards,

Zsofia Freund
